# ALL-BUT-THE-TOP: SIMPLE AND EFFECTIVE POST-PROCESSING FOR WORD REPRESENTATIONS

**Jiaqi Mu, Pramod Viswanath**
University of Illinois at Urbana Champaign
{jiaqimu2, pramodv}@illinois.edu

## ABSTRACT

Real-valued word representations have transformed NLP applications; popular examples are word2vec and GloVe, recognized for their ability to capture linguistic regularities. In this paper, we demonstrate a *very simple*, and yet counter-intuitive, postprocessing technique – eliminate the common mean vector and a few top dominating directions from the word vectors – that renders off-the-shelf representations *even stronger*. The postprocessing is empirically validated on a variety of lexical-level intrinsic tasks (word similarity, concept categorization, word analogy) and sentence-level tasks (semantic textural similarity and text classification) on multiple datasets and with a variety of representation methods and hyperparameter choices in multiple languages; in each case, the processed representations are consistently better than the original ones.

## 1 INTRODUCTION

Words and their interactions (as sentences) are the basic units of natural language. Although words are readily modeled as discrete atomic units, this is unable to capture the relation between the words. Recent distributional real-valued representations of words (examples: word2vec, GloVe) have transformed the landscape of NLP applications – for instance, text classification (Socher et al., 2013b; Maas et al., 2011; Kim, 2014), machine translation (Sutskever et al., 2014; Bahdanau et al., 2014) and knowledge base completion (Bordes et al., 2013; Socher et al., 2013a). The success comes from the geometry of the representations that efficiently captures linguistic regularities: the semantic similarity of words is well captured by the similarity of the corresponding vector representations.

A variety of approaches have been proposed in recent years to learn the word representations: Collobert et al. (2011); Turian et al. (2010) learn the representations via semi-supervised learning by jointly training the language model and downstream applications; Bengio et al. (2003); Mikolov et al. (2010); Huang et al. (2012) do so by fitting the data into a neural network language model; Mikolov et al. (2013); Mnih & Hinton (2007) by log-linear models; and Dhillon et al. (2012); Pennington et al. (2014); Levy & Goldberg (2014); Stratos et al. (2015); Arora et al. (2016) by producing a low-dimensional representation of the cooccurrence statistics. Despite the wide disparity of algorithms to induce word representations, the performance of several of the recent methods is roughly similar on a variety of intrinsic and extrinsic evaluation testbeds.

In this paper, we find that a *simple* processing renders the off-the-shelf existing representations *even stronger*. The proposed algorithm is motivated by the following observation.

**Observation** *Every* representation we tested, in many languages, has the following properties:

- The word representations have *non-zero mean* – indeed, word vectors share a large common vector (with norm up to a half of the average norm of word vector).
- After removing the common mean vector, the representations are *far from* isotropic – indeed, much of the energy of most word vectors is contained in a very low dimensional subspace (say, 8 dimensions out of 300).

**Implication** Since all words share the same common vector and have the same dominating directions, and such vector and directions strongly influence the word representations in the same way, we

propose to eliminate them by: (a) removing the nonzero mean vector from all word vectors, effectively reducing the energy; (b) projecting the representations *away* from the dominating $D$ directions, effectively reducing the dimension. Experiments suggest that $D$ depends on the representations (for example, the dimension of the representation, the training methods and their specific hyperparameters, the training corpus) and also depends on the downstream applications. Nevertheless, a rule of thumb of choosing $D$ around $d/100$, where $d$ is the dimension of the word representations, works uniformly well across multiple languages and multiple representations and multiple test scenarios.

We emphasize that the proposed postprocessing is *counter intuitive* – typically denoising by dimensionality reduction is done by eliminating the *weakest* directions (in a singular value decomposition of the stacked word vectors), and *not* the dominating ones. Yet, such postprocessing yields a "purified" and more "isotropic" word representation as seen in our elaborate experiments.

**Experiments**   By postprocessing the word representation by eliminating the common parts, we find the processed word representations to capture stronger linguistic regularities. We demonstrate this quantitatively, by comparing the performance of both the original word representations and the processed ones on three canonical lexical-level tasks:

- *word similarity* task tests the extent to which the representations capture the similarity between two words – the processed representations are consistently better on seven different datasets, on average by 1.7%;
- *concept categorization* task tests the extent to which the clusters of word representations capture the word semantics – the processed representations are consistently better on three different datasets, by 2.8%, 4.5% and 4.3%;
- *word analogy* task tests the extent to which the difference of two representations captures a latent linguistic relation – again, the performance is consistently improved (by 0.5% on semantic analogies, 0.2% on syntactic analogies and 0.4% in total). Since part of the dominant components are inherently canceled due to the subtraction operation while solving the analogy, we posit that the performance improvement is not as pronounced as earlier.

Extrinsic evaluations provide a way to test the goodness of representations in specific downstream tasks. We evaluate the effect of postprocessing on a standardized and important extrinsic evaluation task on sentence modeling: *semantic textual similarity* task – where we represent a sentence by its averaged word vectors and score the similarity between a pair of sentences by the cosine similarity between the corresponding sentence representation. Postprocessing improves the performance consistently and significantly over 21 different datasets (average improvement of 4%).

Word representations have been particularly successful in NLP applications involving supervised-learning, especially in conjunction with neural network architecture. Indeed, we see the power of postprocessing in an experiment on a standard *text classification* task using a well established convoluntional neural network (CNN) classifier (Kim, 2014) and three RNN classifiers (with vanilla RNN, GRU (Chung et al., 2015) and LSTM Greff et al. (2016) as recurrent units). Across two different pre-trained word vectors, five datasets and four different architectures, the performance with processing improves on a majority of instances (34 out of 40) by a good margin (2.85% on average), and the two performances with and without processing are comparable in the remaining ones.

**Related Work.**   Our work is directly related to word representation algorithms, most of which have been elaborately cited.

Aspects similar to our postprocessing algorithm have appeared in specific NLP contexts very recently in (Sahlgren et al., 2016) (centering the mean) and (Arora et al., 2017) (nulling away only the first principal component). Although there is a superficial similarity between our work and (Arora et al. 2017), the nulling directions we take and the one they take are fundamentally different. Specifically, in Arora et al. (2017), the first dominating vector is *dataset-specific*, i.e., they first compute the sentence representation for the entire semantic textual similarity dataset, then extract the top direction from those sentence representations and finally project the sentence representation away from it. By doing so, the top direction will inherently encode the common information across the entire dataset, the top direction for the "headlines" dataset may encode common information about news articles while the top direction for "Twitter'15" may encode the common information about tweets. In contrast, our dominating vectors are over the entire vocabulary of the language.

More generally, the idea of removing the top principal components has been studied in the context of *positive-valued, high-dimensional* data matrix analysis (Bullinaria & Levy, 2012; Price et al., 2006). Bullinaria & Levy (2012) posits that the highest variance components of the cooccurrence matrix are corrupted by information other than lexical semantics, thus heuristically justifying the removal of the top principal components. A similar idea appears in the context of population matrix analysis (Price et al., 2006), where the entries are also all positive. Our postprocessing operation is on dense low-dimensional representations (with both positive and negative entries).

We posit that the postprocessing operation makes the representations more "isotropic" with stronger self-normalization properties – discussed in detail in Section 2 and Appendix A. Our main point is that this isotropy condition can be explicitly enforced to come up with new embedding algorithms (of which our proposed post-processing is a simple and practical version).

## 2 POSTPROCESSING

We test our observations on various word representations: four publicly available word representations (WORD2VEC[1] (Mikolov et al., 2013) trained using Google News, GLOVE[2] (Pennington et al., 2014) trained using Common Crawl, RAND-WALK (Arora et al., 2016) trained using Wikipedia and TSCCA[3] trained using English Gigaword) and two self-trained word representations using CBOW and Skip-gram (Mikolov et al., 2013) on the 2010 Wikipedia corpus from (Al-Rfou et al., 2013). The detailed statistics for all representations are listed in Table 1. For completeness, we also consider the representations on other languages: a detailed study is provided in Appendix C.2.

|  | Language | Corpus | dim | vocab size | avg. $\|v(w)\|_2$ | $\|\mu\|_2$ |
|---|---|---|---|---|---|---|
| WORD2VEC | English | Google News | 300 | 3,000,000 | 2.04 | 0.69 |
| GLOVE | English | Common Crawl | 300 | 2,196,017 | 8.30 | 3.15 |
| RAND-WALK | English | Wikipedia | 300 | 68, 430 | 2.27 | 0.70 |
| CBOW | English | Wikipedia | 300 | 1,028,961 | 1.14 | 0.29 |
| Skip-Gram | English | Wikipedia | 300 | 1,028,961 | 2.32 | 1.25 |

Table 1: A detailed description for the embeddings in this paper.

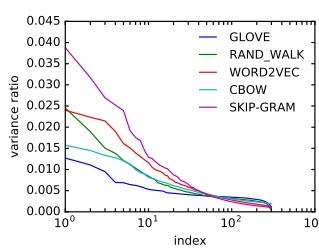

Figure 1: The decay of the normalized singular values of word representation.

Let $v(w) \in \mathbb{R}^d$ be a word representation for a given word $w$ in the vocabulary $\mathcal{V}$. We observe the following two phenomena in each of the word representations listed above:

- $\{v(w) : w \in \mathcal{V}\}$ are not of zero-mean: i.e., all $v(w)$ share a non-zero common vector, $v(w) = \tilde{v}(w) + \mu$, where $\mu$ is the average of all $v(w)$'s, i.e., $\mu = 1/|\mathcal{V}| \sum_{w \in \mathcal{V}} v(w)$. The norm of $\mu$ is approximately 1/6 to 1/2 of the average norm of all $v(w)$ (cf. Table 1).

- $\{\tilde{v}(w) : w \in \mathcal{V}\}$ are not isotropic: Let $u_1, ..., u_d$ be the first to the last components recovered by the principal component analysis (PCA) of $\{\tilde{v}(w) : w \in \mathcal{V}\}$, and $\sigma_1, ..., \sigma_d$ be the corresponding normalized variance ratio. Each $\tilde{v}(w)$ can be written as a linear combinations of $u$: $\tilde{v}(w) = \sum_{i=1}^{d} \alpha_i(w) u_i$. As shown in Figure 1, we observe that $\sigma_i$ decays near exponentially for small values of $i$ and remains roughly constant over the later ones. This suggests there exists $D$ such that $\alpha_i \gg \alpha_j$ for all $i \leq D$ and $j \gg D$; from Figure 1 one observes that $D$ is roughly 10 with dimension $d = 300$.

**Angular Asymmetry of Representations** A modern understanding of word representations involves either PMI-based (including word2vec (Mikolov et al., 2010; Levy & Goldberg, 2014) and GloVe (Pennington et al., 2014)) or CCA-based spectral factorization approaches. While CCA-based

---

[1] https://code.google.com/archive/p/word2vec/
[2] https://github.com/stanfordnlp/GloVe
[3] http://www.pdhillon.com/code.html

spectral factorization methods have long been understood from a probabilistic (i.e., generative model) view point (Browne, 1979; Hotelling, 1936) and recently in the NLP context (Stratos et al., 2015), a corresponding effort for the PMI-based methods has only recently been conducted in an inspired work (Arora et al., 2016).

Arora et al. (2016) propose a generative model (named RAND-WALK) of sentences, where every word is parameterized by a $d$-dimensional vector. With a key postulate that the word vectors are angularly uniform ("isotropic"), the family of PMI-based word representations can be explained under the RAND-WALK model in terms of the maximum likelihood rule. Our observation that word vectors learnt through PMI-based approaches are not of zero-mean and are not isotropic (c.f. Section 2) contradicts with this postulate. The isotropy conditions are relaxed in Section 2.2 of (Arora et al., 2016), but the match with the spectral properties observed in Figure 1 is not immediate.

This contradiction is explicitly resloved by relaxing the constraints on the word vectors to directly fit the observed spectral properties. The relaxed conditions are: the word vectors should be isotropic around a point (whose distance to the origin is a small fraction of the average norm of word vectors) lying on a low dimensional subspace. Our main result is to show that even with this enlarged parameter-space, the maximum likelihood rule continues to be close to the PMI-based spectral factorization methods. A brief summary of RAND-WALK, and the mathematical connection between our work and theirs, are explored in detail in Appendix A.

## 2.1 ALGORITHM

Since all word representations share the same common vector $\mu$ and have the same dominating directions and such vector and directions strongly influence the word representations in the same way, we propose to eliminate them, as formally achieved as Algorithm 1.

---

**Algorithm 1:** Postprocessing algorithm on word representations.

**Input** : Word representations $\{v(w), w \in \mathcal{V}\}$, a threshold parameter $D$,
1 Compute the mean of $\{v(w), w \in \mathcal{V}\}$, $\mu \leftarrow \frac{1}{|\mathcal{V}|} \sum_{w \in \mathcal{V}} v(w), \tilde{v}(w) \leftarrow v(w) - \mu$
2 Compute the PCA components: $u_1, ..., u_d \leftarrow \text{PCA}(\{\tilde{v}(w), w \in \mathcal{V}\})$.
3 Preprocess the representations: $v'(w) \leftarrow \tilde{v}(w) - \sum_{i=1}^{D} \left( u_i^\top v(w) \right) u_i$
**Output :** Processed representations $v'(w)$.

---

**Significance of Nulled Vectors** Consider the representation of the words as viewed in terms of the top $D$ PCA coefficients $\alpha_\ell(w)$, for $1 \le \ell \le D$. We find that these few coefficients encode the *frequency* of the word to a significant degree; Figure 2 illustrates the relation between the $(\alpha_1(w), \alpha_2(w))$ and the unigram probabilty $p(w)$, where the correlation is geometrically visible.

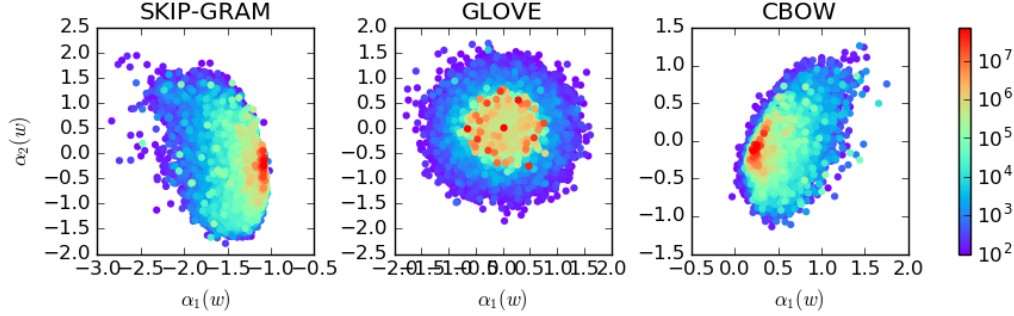

Figure 2: The top two PCA directions (i.e, $\alpha_1(w)$ and $\alpha_2(w)$) encode frequency.

**Discussion** In our proposed processing algorithm, the number of components to be nulled, $D$, is the only hyperparameter that needs to be tuned. We find that a good rule of thumb is to choose $D$ approximately to be $d/100$, where $d$ is the dimension of a word representation. This is empirically justified in the experiments of the following section where $d = 300$ is standard for published word representations. We trained word representations for higher values of $d$ using the WORD2VEC and

GLOVE algorithms and repeated these experiments; we see corresponding consistent improvements due to postprocessing in Appendix C.

## 2.2 POSTPROCESSING AS A "ROUNDING" TOWARDS ISOTROPY

The idea of isotropy comes from the partition function defined in (Arora et al., 2016),

$$Z(c) = \sum_{w \in \mathcal{V}} \exp\left(c^\top v(w)\right),$$

where $Z(c)$ should approximately be a constant with any unit vector $c$ (c.f. Lemma 2.1 in (Arora et al., 2016)). Hence, we mathematically define a measure of isotropy as follows,

$$I(\{v(w)\}) = \frac{\min_{\|c\|=1} Z(c)}{\max_{\|c\|=1} Z(c)}, \tag{1}$$

where $I(\{v(w)\})$ ranges from 0 to 1, and $I(\{v(w)\})$ closer to 1 indicates that $\{v(w)\}$ is more isotropic. The intuition behind our postprocessing algorithm can also be motivated by letting $I(\{v(w)\}) \to 1$.

Let $V$ be the matrix stacked by all word vectors, where the rows correspond to word vectors, and $1_{|\mathcal{V}|}$ be the $|\mathcal{V}|$-dimensional vectors with all entries equal to one, $Z(c)$ can be equivalently defined as follows,

$$Z(c) = |\mathcal{V}| + 1_{|\mathcal{V}|}^\top V c + \frac{1}{2} c^\top V^\top V c + \sum_{k=3}^{\infty} \frac{1}{k!} \sum_{w \in \mathcal{V}} (c^\top v(w))^k.$$

$I(\{v(w)\})$ is, therefore, can be *very coarsely* approximated by,

- **A first order approximation**:

$$I(\{v(w)\}) \approx \frac{|\mathcal{V}| + \min_{\|c\|=1} 1_{|\mathcal{V}|}^\top V c}{|\mathcal{V}| + \max_{\|c\|=1} 1_{|\mathcal{V}|}^\top V c} = \frac{|\mathcal{V}| - \|1_{|\mathcal{V}|}^\top V\|}{|\mathcal{V}| + \|1_{|\mathcal{V}|}^\top V\|}.$$

  Letting $I(\{v(w)\}) = 1$ yields $\|1_{|\mathcal{V}|}^\top V\| = 0$, which is equivalent to $\sum_{w \in \mathcal{V}} v(w) = 0$. The intuition behind the first order approximation matches with the first step of the proposed algorithm, where we enforce $v(w)$ to have a zero mean.

- **A second order approximation**:

$$I(\{v(w)\}) \approx \frac{|\mathcal{V}| + \min_{\|c\|=1} 1_{|\mathcal{V}|}^\top V c + \min_{\|c\|=1} \frac{1}{2} c^\top V^\top V c}{|\mathcal{V}| + \max_{\|c\|=1} 1_{|\mathcal{V}|}^\top V c + \max_{\|c\|=1} \frac{1}{2} c^\top V^\top V c} = \frac{|\mathcal{V}| - \|1_{|\mathcal{V}|}^\top V\| + \frac{1}{2}\sigma_{\min}^2}{|\mathcal{V}| + \|1_{|\mathcal{V}|}^\top V\| + \frac{1}{2}\sigma_{\max}^2},$$

  where $\sigma_{\min}$ and $\sigma_{\max}$ are the smallest and largest singular value of $V$, respectively. Letting $I(\{v(w)\}) = 1$ yields $\|1_{|\mathcal{V}|}^\top V\| = 0$ and $\sigma_{\min} = \sigma_{\max}$. The fact that $\sigma_{\min} = \sigma_{\max}$ suggests the spectrum of $v(w)$'s should be flat. The second step of the proposed algorithm removes the highest singular values, and therefore explicitly flatten the spectrum of $V$.

**Empirical Verification**  Indeed, we empirically validate the effect of postprocessing of on $I(\{v(w)\})$. Since there is no closed-form solution for $\arg\max_{\|c\|=1} Z(c)$ or $\arg\min_{\|c\|=1} Z(c)$, and it is impossible to enumerate all $c$'s, we estimate the measure by,

$$I(\{v(w)\}) \approx \frac{\min_{c \in C} Z(c)}{\max_{c \in C} Z(c)},$$

where $C$ is the set of eigenvectors of $V^\top V$. The value of $I(\{v(w)\})$ for the original vectors and processed ones are reported in Table 2, where we can observe that the degree of isotropy vastly increases in terms of this measure.

|  | before | after |
|---|---|---|
| WORD2VEC | 0.7 | **0.95** |
| GLOVE | 0.065 | **0.6** |

Table 2: Before-After on the measure of isotropy.

A formal way to verify the isotropy property is to directly check if the "self-normalization" property (i.e., $Z(c)$ is a constant, independent of $c$ (Andreas & Klein, 2015)) holds more strongly. Such a validation is seen diagrammatically in Figure 3 where we randomly sampled 1,000 $c$'s as (Arora et al., 2016)

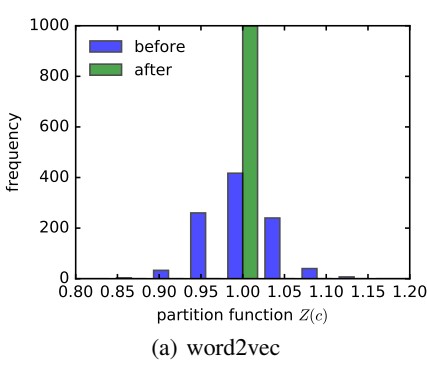
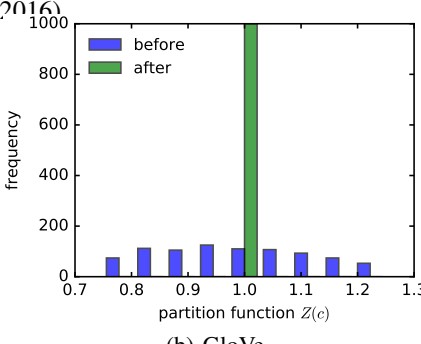

(a) word2vec  (b) GloVe

Figure 3: The histogram of $Z(c)$ for 1,000 randomly sampled vectors $c$ of unit norm, where $x$-axis is normalized by the mean of all values and $D = 2$ for GLOVE and $D = 3$ for WORD2VEC.

## 3 EXPERIMENTS

Given the popularity and widespread use of WORD2VEC (Mikolov et al., 2013) and GLOVE (Pennington et al., 2014), we use their publicly available pre-trained reprepsentations in the following experiments. We choose $D = 3$ for WORD2VEC and $D = 2$ for GLOVE. The key underlying principle behind word representations is that similar words should have similar representations. Following the tradition of evaluating word representations (Schnabel et al., 2015; Baroni et al., 2014), we perform three canonical *lexical-level* tasks: (a) word similarity; (b) concept categorization; (c) word analogy; and one sentence-level task: (d) semantic textual similarity. The processed representations consistently improve performance on all three of them, and especially strongly on the first two.

|  | WORD2VEC | | GLOVE | |
|---|---|---|---|---|
|  | orig. | proc. | orig. | proc. |
| RG65 | 76.08 | **78.34** | **76.96** | 74.36 |
| WS | 68.29 | **69.05** | 73.79 | **76.79** |
| RW | 53.74 | **54.33** | 46.41 | **52.04** |
| MEN | 78.20 | **79.08** | 80.49 | **81.78** |
| MTurk | 68.23 | **69.35** | 69.29 | **70.85** |
| SimLex | 44.20 | **45.10** | 40 83 | **44.97** |
| SimVerb | 36.35 | **36.50** | 28.33 | **32.23** |

Table 3: Before-After results (x100) on word similarity task on seven datasets.

**Word Similarity** The word similarity task is as follows: given a pair of words, the algorithm assigns a "similarity" score – if the pair of words are highly related then the score should also be high and vice versa. The algorithm is evaluated in terms of Spearman's rank correlation compared to (a gold set of) human judgements.

For this experiment, we use seven standard datasets: the first published RG65 dataset (Rubenstein & Goodenough, 1965); the widely used WordSim-353 (WS) dataset (Finkelstein et al., 2001) which contains 353 pairs of commonly used verbs and nouns; the rare-words (RW) dataset (Luong et al., 2013) composed of rarely used words; the MEN dataset (Bruni et al., 2014) where the 3000 pairs of words are rated by crowdsourced participants; the MTurk dataset (Radinsky et al., 2011) where the 287 pairs of words are rated in terms of relatedness; the SimLex-999 (SimLex) dataset (Hill et al., 2016) where the score measures "genuine" similarity; and lastly the SimVerb-3500 (SimVerb) dataset (Gerz et al., 2016), a newly released large dataset focusing on similarity of verbs.

In our experiment, the algorithm scores the similarity between two words by the cosine similarity between the two corresponding word vectors ($\mathrm{CosSim}(v_1, v_2) = v_1^\top v_2 / \|v_1\|\|v_2\|$). The detailed performance on the seven datasets is reported in Table 3, where we see a consistent and significant performance improvement due to postprocessing, across all seven datasets. These statistics

(average improvement of **2.3**%) suggest that by removing the common parts, the remaining word representations are able to capture stronger semantic relatedness/similarity between words.

|  | WORD2VEC | | GLOVE | |
|---|---|---|---|---|
|  | orig. | proc. | orig. | proc. |
| ap | 54.43 | **57.72** | 64 .18 | **65.42** |
| esslli | 75.00 | **84.09** | 81.82 | 81.82 |
| battig | 71.97 | **81.71** | 86.59 | 86.59 |

Table 4: Before-After results (x100) on the categorization task.

**Concept Categorization** This task is an indirect evaluation of the similarity principle: given a set of concepts, the algorithm needs to group them into different categories (for example, "bear" and "cat" are both animals and "city" and "country" are both related to districts). The clustering performance is then evaluated in terms of purity (Manning et al., 2008) – the fraction of the total number of the objects that were classified correctly.

We conduct this task on three different datasets: the Almuhareb-Poesio (ap) dataset (Almuhareb, 2006) contains 402 concepts which fall into 21 categories; the ESSLLI 2008 Distributional Semantic Workshop shared-task dataset (Baroni et al., 2008) that contains 44 concepts in 6 categories; and the Battig test set (Baroni & Lenci, 2010) that contains 83 words in 10 categories.

Here we follow the setting and the proposed algorithm in (Baroni et al., 2014; Schnabel et al., 2015) – we cluster words (via their representations) using the classical $k$-Means algorithm (with fixed $k$). Again, the processed vectors perform consistently better on all three datasets (with average improvement of 2.5%); the full details are in Table 4.

|  | WORD2VEC | | GLOVE | |
|---|---|---|---|---|
|  | orig. | proc. | orig. | proc. |
| syntax | 73.46 | **73.50** | 74.95 | **75.40** |
| semantics | 72.28 | **73.36** | 79.22 | **79.25** |
| all | 72.93 | **73.44** | 76.89 | **77.15** |

Table 5: Before-After results (x100) on the word analogy task.

**Word Analogy** The analogy task tests to what extent the word representations can encode latent linguistic relations between a pair of words. Given three words $w_1$, $w_2$, and $w_3$, the analogy task requires the algorithm to find the word $w_4$ such that $w_4$ is to $w_3$ as $w_2$ is to $w_1$.

We use the analogy dataset introduced in (Mikolov et al., 2013). The dataset can be divided into two parts: (a) the *semantic* part containing around 9k questions, focusing on the latent semantic relation between pairs of words (for example, what is to Chicago as Texas is to Houston); and (b) the *syntatic* one containing roughly 10.5k questions, focusing on the latent syntatic relation between pairs of words (for example, what is to "amazing" as "apprently" is to "apparent").

In our setting, we use the original algorithm introduced in (Mikolov et al., 2013) to solve this problem, i.e., $w_4$ is the word that maximize the cosine similarity between $v(w_4)$ and $v(w_2) - v(w_1) + v(w_3)$. The average performance on the analogy task is provided in Table 5 (with a detailed performance provided in Table 19 in Appendix D). It can be noticed that while postprocessing continues to improve the performance, the improvement is not as pronounced as earlier. We hypothesize that this is because the mean and some dominant components get canceled during the subtraction of $v(w_2)$ from $v(w_1)$, and therefore the effect of postprocessing is less relevant.

|  | WORD2VEC | | GLOVE | |
|---|---|---|---|---|
|  | orig. | proc. | orig. | proc. |
| 2012 | 57.22 | **57.67** | 48.27 | **54.06** |
| 2013 | 56.81 | **57.98** | 44.83 | **57.71** |
| 2014 | 62.89 | **63.30** | 51.11 | **59.23** |
| 2015 | 62.74 | **63.35** | 47.23 | **57.29** |
| SICK | 70.10 | **70 20** | 65.14 | **67.85** |
| all | 60.88 | **61.45** | 49.19 | **56.76** |

Table 6: Before-After results (x100) on the semantic textual similarity tasks.

**Semantic Textual Similarity** Extrinsic evaluations measure the contribution of a word representation to specific downstream tasks; below, we study the effect of postprocessing on a standard sentence modeling task – *semantic textual similarity* (STS) which aims at testing the degree to which the algorithm can capture the semantic equivalence between two sentences. For each pair of sentences, the algorithm needs to measure how similar the two sentences are. The degree to which the measure matches with human judgment (in terms of Pearson correlation) is an index of the algorithm's performance. We test the word representations on 20 textual similarity datasets

from the 2012-2015 SemEval STS tasks (Agirre et al., 2012; 2013; 2014; 2015), and the 2012 SemEval Semantic Related task (SICK) (Marelli et al., 2014).

Representing sentences by the average of their constituent word representations is surprisingly effective in encoding the semantic information of sentences (Wieting et al., 2015; Adi et al., 2016) and close to the state-of-the-art in these datasets. We follow this rubric and represent a sentence $s$ based on its averaged word representation, i.e., $v(s) = \frac{1}{|s|} \sum_{w \in s} v(w)$, and then compute the similarity between two sentences via the cosine similarity between the two representations. The average performance of the original and processed representations is itemized in Table 6 (with a detailed performance in Table 20 in Appendix E) – we see a consistent and significant improvement in performance because of postprocessing (on average **4**% improvement).

## 4 POSTPROCESSING AND SUPERVISED CLASSIFICATION

Supervised downstream NLP applications have greatly improved their performances in recent years by combining the discriminative learning powers of neural networks in conjunction with the word representations. We evaluate the performance of a variety of neural network architectures on a standard and important NLP application: *text classification*, with sentiment analysis being a particularly important and popular example. The task is defined as follows: given a sentence, the algorithm needs to decide which category it falls into. The categories can be either binary (e.g., positive/negative) or can be more fine-grained (e.g. very positive, positive, neutral, negative, and very negative).

We evaluate the word representations (with and without postprocessing) using four different neural network architectures (CNN, vanilla-RNN, GRU-RNN and LSTM-RNN) on five benchmarks: (a) the movie review (MR) dataset (Pang & Lee, 2005); (b) the subjectivity (SUBJ) dataset (Pang & Lee, 2004); (c) the TREC question dataset (Li & Roth, 2002); (d) the IMDb dataset (Maas et al., 2011); (e) the stanford sentiment treebank (SST) dataset (Socher et al., 2013a). A detailed description of these standard datasets, their training/test parameters and the cross validation methods adopted is in Appendix F. Specifically, we allow the parameter $D$ (i.e., the number of nulled components) to vary between 0 and 4, and the best performance of the four neural network architectures with the now-standard CNN-based text classification algorithm (Kim, 2014) (implemented using tensorflow[4]) is itemized in Table 7. The key observation is that the performance of postprocessing is better in a majority (34 out of 40) of the instances by 2.32% on average, and in the rest the instances the two performances (with and without postprocessing) are comparable.

| | CNN | | | | vanilla-RNN | | | | GRU-RNN | | | | LSTM-RNN | | | |
|---|---|---|---|---|---|---|---|---|---|---|---|---|---|---|---|---|
| | WORD2VEC | | GLOVE | | WORD2VEC | | GLOVE | | WORD2VEC | | GLOVE | | WORD2VEC | | GLOVE | |
| | orig. | proc. | orig. | proc. | orig. | proc. | orig. | proc. | orig. | proc. | orig. | proc. | orig. | proc. | orig. | proc. |
| MR | 70.80 | **71.27** | 71.01 | **71.11** | **74.95** | 74.01 | 71.14 | **72.56** | 77.86 | **78.26** | 74.98 | **75.13** | 75.69 | **77.34** | **72.02** | 71.84 |
| SUBJ | 87.14 | **87.33** | 86.98 | **87.25** | 82.85 | **87.60** | 81.45 | **87.37** | 90.96 | **91.10** | 91.16 | **91.85** | 90.23 | **90.54** | 90.74 | **90.82** |
| TREC | 87.80 | **89.00** | 87.60 | **89.00** | 80.60 | **89.20** | 85.20 | **89.00** | 91.60 | **92.40** | 91.60 | **93.00** | 88.00 | **91.20** | 85.80 | **91.20** |
| SST | **38.46** | 38.33 | **38.82** | 37.83 | **42.08** | 39.91 | 41.45 | **41.90** | 41.86 | **45.02** | 36.52 | **37.69** | **43.08** | 42.08 | 37.51 | **38.05** |
| IMDb | 86.68 | **87.12** | **87.27** | 87.10 | 50.15 | **53.14** | 52.76 | **76.07** | 82.96 | **83.47** | 81.50 | **82.44** | 81.29 | **82.60** | 79.10 | **81.33** |

Table 7: Before-After results (x100) on the text classification task using CNN (Kim, 2014) and vanilla RNN, GRU-RNN and LSTM-RNN.

A further validation of the postprocessing operation in a variety of downstream applications (eg: named entity recognition, syntactic parsers, machine translation) and classification methods (eg: random forests, neural network architectures) is of active research interest. Of particular interest is the impact of the postprocessing on the rate of convergence and generalization capabilities of the classifiers. Such a systematic study would entail a concerted and large-scale effort by the research community and is left to future research.

**Discussion** All neural network architectures, ranging from feedforward to recurrent (either vanilla or GRU or LSTM), implement at least linear processing of hidden/input state vectors at each of their nodes; thus the postprocessing operation suggested in this paper can *in principle* be automatically "learnt" by the neural network, if such internal learning is in-line with the end-to-end training examples. Yet, in practice this is complicated due to limitations of optimization procedures (SGD) and sample

---

[4]https://github.com/dennybritz/cnn-text-classification-tf

noise. We conduct a preliminary experiment in Appendix B and show that subtracting the mean (i.e., the first step of postprocessing) is "effectively learnt" by neural networks within their nodes.

## 5 CONCLUSION

We present a simple postprocessing operation that renders word representations even stronger, by eliminating the top principal components of all words. Such an simple operation could be used for word embeddings in downstream tasks or as intializations for training task-specific embeddings. Due to their popularity, we have used the published representations of WORD2VEC and GLOVE in English in the main text of this paper; postprocessing continues to be successful for other representations and in multilingual settings – the detailed empirical results are tabulated in Appendix C.

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

# Appendix: All-but-the-Top: Simple and Effective postprocessing for Word Representations

## A ANGULAR ASYMMETRY OF REPRESENTATIONS

A modern understanding of word representations involves either PMI-based (including word2vec (Mikolov et al., 2010; Levy & Goldberg, 2014) and GloVe (Pennington et al., 2014)) or CCA-based spectral factorization approaches. While CCA-based spectral factorization methods have long been understood from a probabilistic (i.e., generative model) view point (Browne, 1979; Hotelling, 1936) and recently in the NLP context (Stratos et al., 2015), a corresponding effort for the PMI-based methods has only recently been conducted in an inspired work (Arora et al., 2016).

(Arora et al., 2016) propose a generative model (named RAND-WALK) of sentences, where every word is parameterized by a $d$-dimensional vector. With a key postulate that the word vectors are angularly uniform ("isotropic"), the family of PMI-based word representations can be explained under the RAND-WALK model in terms of the maximum likelihood rule. Our observation that word vectors learnt through PMI-based approaches are not of zero-mean and are not isotropic (c.f. Section 2) contradicts with this postulate. The isotropy conditions are relaxed in Section 2.2 of (Arora et al., 2016), but the match with the spectral properties observed in Figure 1 is not immediate.

In this section, we resolve this by explicitly relaxing the constraints on the word vectors to directly fit the observed spectral properties. The relaxed conditions are: the word vectors should be isotropic around a point (whose distance to the origin is a small fraction of the average norm of word vectors) lying on a low dimensional subspace. Our main result is to show that even with this enlarged parameter-space, the maximum likelihood rule continues to be close to the PMI-based spectral factorization methods. Formally, the model, the original constraints of (Arora et al., 2016) and the enlarged constraints on the word vectors are listed below:

- **A generative model of sentences**: the word at time $t$, denoted by $w_t$, is generated via a log-linear model with a latent discourse variable $c_t$ (Arora et al., 2016), i.e.,

$$p(w_t|c_t) = \frac{1}{Z(c_t)} \exp\left(c_t^\top v(w_t)\right),\tag{2}$$

  where $v(w) \in \mathbb{R}^d$ is the vector representation for a word $w$ in the vocabulary $V$, $c_t$ is the latent variable which forms a "slowly moving" random walk, and the partition function: $Z(c) = \sum_{w \in \mathcal{V}} \exp\left(c^\top v(w)\right)$.

- **Constraints on the word vectors**: (Arora et al., 2016) suppose that there is a Bayesian priori on the word vectors:

  The ensemble of word vectors consists of i.i.d. draws generated by $v = s \cdot \hat{v}$, where $\hat{v}$ is from the spherical Gaussian distribution, and $s$ is a scalar random variable.

  A deterministic version of this prior is discussed in Section 2.2 of (Arora et al., 2016), but part of these (relaxed) conditions on the word vectors are specifically meant for Theorem 4.1 and not the main theorem (Theorem 2.2). The geometry of the word representations is only evaluated via the ratio of the quadratic *mean* of the singular values to the smallest one being small enough. This meets the relaxed conditions, but not sufficient to validate the proof approach of the main result (Theorem 2.2); what would be needed is that the ratio of the *largest* singular value to the smallest one be small.

- **Revised conditions**: We revise the Bayesian prior postulate (and in a deterministic fashion) formally as follows: there is a mean vector $\mu$, $D$ *orthonormal* vectors $u_1, \ldots, u_D$ (that are orthogonal and of unit norm), such that every word vector $v(w)$ can be represented by,

$$v(w) = \mu + \sum_{i=1}^{D} \alpha_i(w)u_i + \tilde{v}(w),\tag{3}$$

  where $\mu$ is bounded, $\alpha_i$ is bounded by $A$, $D$ is bounded by $DA^2 = o(d)$, $\tilde{v}(w)$ are statistically isotropic. By statistical isotropy, we mean: for high-dimensional rectangles $R$,

$\frac{1}{|\mathcal{V}|} \sum_{w \in \mathcal{V}} \mathbf{1}(\tilde{v}(w) \in R) \to \int_R f(\tilde{v}) d\tilde{v}$, as $|\mathcal{V}| \to \infty$, where $f$ is an angle-independent pdf, i.e., $f(\tilde{v})$ is a function of $\|\tilde{v}\|$.

The revised postulate differs from the original one in two ways: (a) it imposes a formal deterministic constraint on the word vectors; (b) the revised postulate allows the word vectors to be angularly asymmetric: as long as the energy in the direction of $u_1, \ldots, u_D$ is bounded, there is no constraint on the coefficients. Indeed, note that there is no constraint on $\tilde{v}(w)$ to be orthogonal to $u_1, \ldots u_D$.

**Empirical Validation**    We can verify that the enlarged conditions are met by the existing word representations. Specifically, the natural choice for $\mu$ is the mean of the word representations and $u_1 \ldots u_D$ are the singular vectors associated with the top $D$ singular values of the matrix of word vectors. We pick $D = 20$ for WORD2VEC and $D = 10$ for GLOVE, and the corresponding value of $DA^2$ for WORD2VEC and GLOVE vectors are both roughly 40, respectively; both values are small compared to $d = 300$.

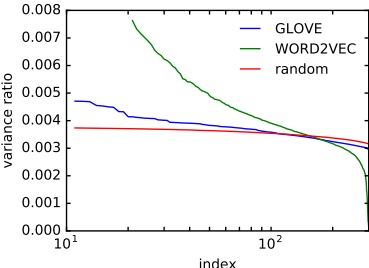

Figure 4: Spectrum of the published WORD2VEC and GLOVE and random Gaussian matrices, ignoring the top $D$ components; $D = 10$ for GLOVE and $D = 20$ for WORD2VEC.

This leaves us to check the statistical isotropy of the "remaining" vectors $\tilde{v}(w)$ for words $w$ in the vocabulary. We do this by plotting the remaining spectrum (i.e. the $(D + 1)$-th, ..., 300th singular values) for the published WORD2VEC and GLOVE vectors in Figure 4. As a comparison, the empirical spectrum of a random Gaussian matrix is also plotted in Figure 4. We see that both spectra are flat (since the vocabulary size is much larger than the dimension $d = 300$). Thus the postprocessing operation can also be viewed as a way of making the vectors "more isotropic".

**Mathematical Contribution**    Under the revised postulate, we show that the main theorem in (Arora et al., 2016) (c.f. Theorem 2.2) still holds. Formally:

**Theorem A.1** *Suppose the word vectors satisfy the constraints. Then*

$$\text{PMI}(w_1, w_2) \stackrel{\text{def}}{=} \log \frac{p(w_1, w_2)}{p(w_1)p(w_2)} \to \frac{v(w_1)^\top v(w_2)}{d}, \quad as \ |\mathcal{V}| \to \infty, \tag{4}$$

*where $p(w)$ is the unigram distribution induced from the model (2), and $p(w_1, w_2)$ is the probability that two words $w_1$ and $w_2$ occur with each other within distance q.*

The proof is in Appendix G. Theorem A.1 suggests that the RAND-WALK generative model and its properties proposed by (Arora et al., 2016) can be generalized to a broader setting (with a relaxed restriction on the geometry of word representations) – relevantly, this relaxation on the geometry of word representations is empirically satisfied by the vectors learnt as part of the maximum likelihood rule.

## B    NEURAL NETWORKS LEARN TO POSTPROCESS

Every neural network family posseses the ability to conduct linear processing inside their nodes; this includes feedforward and recurrent and convolutional neural network models. Thus, in principle, the postprocessing operation can be "learnt and implemented" within the parameters of the neural network. On the other hand, due to the large number of parameters within the neural network, it

is unclear how to verify such a process, even if it were learnt (only one of the layers might be implementing the postprocessing operation or via a combination of multiple effects).

To address this issue, we have adopted a *comparative* approach in the rest of this section. The comparative approach involves adding an *extra layer* interposed in between the inputs (which are word vectors) and the rest of the neural network. This extra layer involves only linear processing. Next we compare the results of the final parameters of the extra layer (trained jointly with the rest of tne neural network parameters, using the end-to-end training examples) with and without preprocessing of the word vectors. Such a comparative approach allows us to separate the effect of the postprocessing operation on the word vectors from the complicated "semantics" of the neural network parameters.

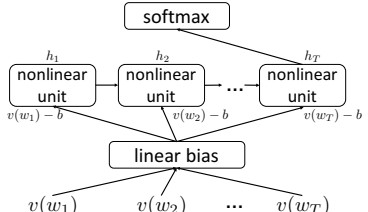

Figure 5: Time-expanded RNN architecture with an appended layer involving linear bias.

|  | vanilla | | GRU | | LSTM | |
|---|---|---|---|---|---|---|
|  | W. | G. | W. | G. | W. | G. |
| MR | 82.07 | 49.23 | 81.35 | 47.63 | 77.95 | 44.92 |
| SUBJ | 84.02 | 49.94 | 83.15 | 50.60 | 83.39 | 48.95 |
| TREC | 81.68 | 52.99 | 82.68 | 50.42 | 80.80 | 46.77 |
| SST | 79.64 | 46.59 | 78.06 | 43.21 | 77.72 | 42.82 |
| IMDb | 93.48 | 66.37 | 94.49 | 55.24 | 87.27 | 46.74 |

Figure 6: The cosine similarity (x100) between $b_{\text{proc.}} + \mu$ and $b_{\text{orig.}}$, where W. and G. stand for WORD2VEC and GLOVE respectively.

**Experiment** We construct a modified neural network by explicitly adding a "postprocessing unit" as the first layer of the RNN architecture (as in Figure 5, where the appended layer is used to test the first step (i.e., remove the mean vector)) of the postprocessing algorithm.

In the modified neural network, the input word vectors are now $v(w) - b$ instead of $v(w)$. Here $b$ is a bias vector trained *jointly* with the rest of the neural network parameters. Note that this is only a relabeling of the parameters from the perspective of the RNN architecture: the nonlinear activation function of the node is now operated on $A(v(w) - b) + b' = Av(w) + (b' - Ab)$ instead of the previous $Av(w) + b'$. Let $b_{\text{proc.}}$ and $b_{\text{orig.}}$ be the inferred biases when using the processed and original word representations, respectively.

We itemize the cosine similarity between $b_{\text{proc.}} + \mu$ and $b_{\text{orig.}}$ in Table 6 for the 5 different datasets and 3 different neural network architectures. In each case, the cosine similarity is remarkably large (on average 0.66, in 300 dimensions) – in other words, trained neural networks implicitly postprocess the word vectors nearly exactly as we proposed. This agenda is successfully implemented in the context of verifying the removal of the mean vector.

The second step of our postprocessing algorithm (i.e., nulling away from the top principal components) is equivalent to applying a projection matrix $P = I - \sum_{i=1}^{D} u_i u_i^\top$ on the word vectors, where $u_i$ is the $i$-th principal component and $D$ is the number of the removed directions. A comparative analysis effort for the second step (nulling the dominant PCA directions) is the following. Instead of applying a bias term $b$, we multiply by a matrix $Q$ to simulate the projection operation. The input word vectors are now $Q_{\text{orig.}} v(w)$ instead of $v(w)$ for the original word vectors, and $Q_{\text{proc.}} P v(w)$ instead of $P v(w)$ for the processed vectors. Testing the similarity between $Q_{\text{orig.}} P$ and $Q_{\text{proc.}}$, allows us to verify if the neural network learns to conduct the projection operation as proposed.

In our experiment, we found that such a result cannot be inferred. One possibility is that there are too many parameters in both $Q_{\text{proc.}}$ and $Q_{\text{orig.}}$, which adds randomness to the experiment. Alternatively, the neural network weights may not be able to learn the second step of the postprocessing operation (indeed, in our experiments postprocessing significantly boosted end-performance of neural network architectures). A more careful experimental setup to test whether the second step of the postprocessing operation is learnt is left as a future research direction.

## C  EXPERIMENTS ON VARIOUS REPRESENTATIONS

In the main text, we have reported empirical results for two published word representations: WORD2VEC and GLOVE, each in 300 dimensions. In this section, we report the results of the same experiments in a variety of other settings to show the generalization capability of the postprocessing operation: representations trained via WORD2VEC and GLOVE algorithms in dimensions other than 300, other representations algorithms (specifically TSCCA and RAND-WALK) and in multiple languages.

### C.1  STATISTICS OF MULTILINGUAL WORD REPRESENTATIONS

We use the publicly available TSCCA representations (Dhillon et al., 2012) on German, French, Spanish, Italian, Dutch and Chinese. The detailed statistics can be found in Table 8 and the decay of their singular values are plotted in Figure 7.

|          | Language | Corpus        | dim | vocab size | avg. $\|v(w)\|_2$ | $\|\mu\|_2$ |
|----------|----------|---------------|-----|------------|-------------------|-------------|
| TSCCA-En | English  | Gigawords     | 200 | 300,000    | 4.38              | 0.78        |
| TSCCA-De | German   | Newswire      | 200 | 300,000    | 4.52              | 0.79        |
| TSCCA-Fr | French   | Gigaword      | 200 | 300,000    | 4.34              | 0.81        |
| TSCCA-Es | Spanish  | Gigaword      | 200 | 300,000    | 4.17              | 0.79        |
| TSCCA-It | Italian  | Newswire+Wiki | 200 | 300,000    | 4.34              | 0.79        |
| TSCCA-Nl | Dutch    | Newswire+Wiki | 200 | 300,000    | 4.46              | 0.72        |
| TSCCA-Zh | Chinese  | Gigaword      | 200 | 300,000    | 4.51              | 0.89        |

Table 8: A detailed description for the TSCCA embeddings in this paper.

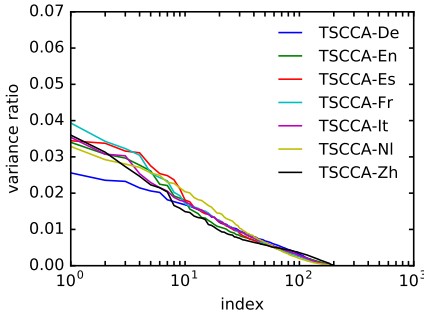

Figure 7: The decay of the normalized singular values of multilingual word representation.

### C.2  MULTILINGUAL GENERALIZATION

In this section, we perform the word similarity task with the original and the processed TSCCA word representations in German and Spanish on three German similarity datasets (GUR65 – a German version of the RG65 dataset, GUR350, and ZG222 in terms of relatedness) (Zesch & Gurevych, 2006) and the Spanish version of RG65 dataset (Camacho-Collados et al., 2015). The choice of $D$ is 2 for both German and Spanish.

The experiment setup and the similarity scoring algorithm are the same as those in Section 3. The detailed experiment results are provided in Table 9, from which we observe that the processed representations are consistently better than the original ones. This provides evidence to the generalization capabilities of the postprocessing operation to multiple languages (similarity datasets in Spanish and German were the only ones we could locate).

### C.3  GENERALIZATION TO DIFFERENT REPRESENTATION ALGORITHMS

Given the popularity and widespread use of WORD2VEC (Mikolov et al., 2013) and GLOVE (Pennington et al., 2014), the main text has solely focused on their published publicly avalable

|        | language | TSCCA | |
|--------|----------|-------|------|
|        |          | orig. | proc. |
| RG65   | Spanish  | 60.33 | **60.37** |
| GUR65  | German   | 61.75 | **64.39** |
| GUR350 | German   | 44.91 | **46.59** |
| ZG222  | German   | 30.37 | **32.92** |

Table 9: Before-After results (x100) on the word similarity task in multiple languages.

300-dimension representations. In this section, we show that the proposed postprocessing algorithm generalizes to other representation methods. Specifically, we demonstrate this on RAND-WALK (obtained via personal communication) and TSCCA (publicly available) on all the experiments of Section 3. The choice of $D$ is 2 for both RAND-WALK and TSCCA.

In summary, the performance improvements on the similarity task, the concept categorization task, the analogy task, and the semantic textual similarity dataset are on average 2.23%, 2.39%, 0.11% and 0.61%, respectively. The detailed statistics are provided in Table 10, Table 11, Table 12 and Table 13, respectively. These results are a testament to the generalization capabilities of the postprocessing algorithm to other representation algorithms.

|         | RAND-WALK | | TSCCA | |
|---------|-------|-------|-------|-------|
|         | orig. | proc. | orig. | proc. |
| RG65    | 80.66 | **82.96** | 47.53 | **47.67** |
| WS      | 65.89 | **74.37** | 54.21 | **54.35** |
| RW      | 45.11 | **51.23** | **43.96** | 43.72 |
| MEN     | 73.56 | **77.22** | 65.48 | **65.62** |
| MTurk   | 64.35 | **66.11** | 59.65 | **60.03** |
| SimLex  | 34.05 | **36.55** | 34.86 | **34.91** |
| SimVerb | 16.05 | **21.84** | 23.79 | **23.83** |

Table 10: Before-After results (x100) on the word similarity task on seven datasets.

|        | RAND-WALK | | TSCCA | |
|--------|-------|-------|-------|-------|
|        | orig. | proc. | orig. | proc. |
| ap     | 59.83 | **62.36** | 60.00 | **63.42** |
| esslli | 72.73 | 72.73 | 68.18 | **70.45** |
| battig | 75.73 | **81.82** | 70.73 | 70.73 |

Table 11: Before-After results (x100) on the categorization task.

|      | RAND-WALK | | TSCCA | |
|------|-------|-------|-------|-------|
|      | orig. | proc. | orig. | proc. |
| syn. | 60.39 | **60.48** | 37.72 | **37.80** |
| sem. | 83.55 | **83.82** | 14.54 | **14.55** |
| all  | 70.50 | **70.67** | 27.30 | **27.35** |

Table 12: Before-After results (x100) on the word analogy task.

## C.4 ROLE OF DIMENSIONS

The main text has focused on the dimension choice of $d = 300$, due to its popularity. In this section we explore the role of the dimension in terms of both choice of $D$ and the performance of the postprocessing operation – we do this by using skip-gram model on the 2010 snapshot of Wikipedia corpus (Al-Rfou et al., 2013) to train word representations. We first observe that the two phenomena of Section 2 continue to hold:

- From Table 14 we observe that the ratio between the norm of $\mu$ and the norm average of all $v(w)$ spans from 1/3 to 1/4;

|  | RAND-WALK | | TSCCA | |
|---|---|---|---|---|
|  | orig. | proc. | orig. | proc. |
| 2012 | **38.03** | 37.66 | 44.51 | **44.63** |
| 2013 | **37.47** | 36.85 | **43.21** | 42.74 |
| 2014 | 46.06 | **48.32** | 52.85 | **52.87** |
| 2015 | 47.82 | **51.76** | **56.22** | 56.14 |
| SICK | 51.58 | **51.76** | **56.15** | 56.11 |
| all | 43.48 | **44.67** | 50.01 | **50.23** |

Table 13: Before-After results (x100) on the semantic textual similarity tasks.

| dim | 300 | 400 | 500 | 600 | 700 | 800 | 900 | 1000 |
|---|---|---|---|---|---|---|---|---|
| avg. $\|v(w)\|_2$ | 4.51 | 5.17 | 5.91 | 6.22 | 6.49 | 6.73 | 6.95 | 7.15 |
| $\|\mu\|_2$ | 1.74 | 1.76 | 1.77 | 1.78 | 1.79 | 1.80 | 1.81 | 1.83 |

Table 14: Statistics on word representation of dimensions 300, 400, ..., and 1000 using the skip-gram model.

- From Figure 8 we observe that the decay of the variance ratios $\sigma_i$ is near exponential for small values of $i$ and remains roughly constant over the later ones.

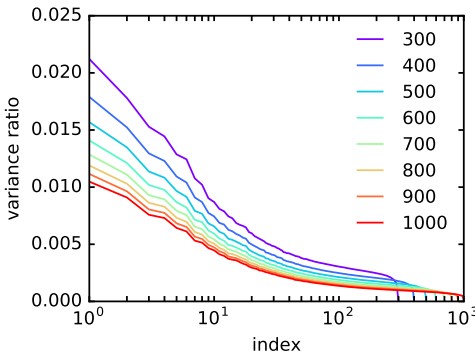

Figure 8: The decay of the normalized singular values of word representations.

A rule of thumb choice of $D$ is around $d/100$. We validate this claim empirically by performing the tasks in Section 3 on word representations of higher dimensions, ranging from 300 to 1000, where we set the parameter $D = d/100$. In summary, the performance improvement on the four itemized tasks of Section 3 are 2.27%, 3.37%, 0.01 and 1.92% respectively; the detailed results can be found in Table 15, Table 16, Table 17, and Table 18. Again, note that the improvement for analogy tasks is marginal. These experimental results justify the rule-of-thumb setting of $D = d/100$, although we emphasize that the improvements can be further accentuated by tuning the choice of $D$ based on the specific setting.

# D  EXPERIMENTS ON WORD ANALOGY TASK

The detailed performance on the analogy task is provided in Table 19.

# E  EXPERIMENTS ON SEMANTIC TEXTUAL SIMILARITY TASK

The detailed performance on the semantic textual similarity is provided in Table 20.

| Dim | 300 | | 400 | | 500 | | 600 | |
|---|---|---|---|---|---|---|---|---|
| | orig. | proc. | orig. | proc. | orig. | proc. | orig. | proc. |
| RG65 | 73.57 | **74.72** | 75.64 | **79.87** | 77.72 | **81.97** | 77.59 | **80.7** |
| WS | 70.25 | **71.95** | 70.8 | **72.88** | 70.39 | **72.73** | 71.64 | **74.04** |
| RW | 46.25 | **49.11** | 45.97 | **47.63** | 46.6 | **48.59** | 45.7 | **47.81** |
| MEN | 75.66 | **77.59** | 76.07 | **77.89** | 75.9 | **78.15** | 75.88 | **78.15** |
| Mturk | 75.66 | **77.59** | 67.68 | **68.11** | 66.89 | **68.25** | 67.6 | **67.87** |
| SimLex | 34.02 | **36.19** | 35.17 | **37.1** | 35.73 | **37.65** | 35.76 | **38.04** |
| SimVerb | 22.22 | **24.98** | 22.91 | **25.32** | 23.03 | **25.82** | 23.35 | **25.97** |
| Dim | 700 | | 800 | | 900 | | 1000 | |
| | orig. | proc. | orig. | proc. | orig. | proc. | orig. | proc. |
| RG65 | 77.3 | **81.07** | 77.52 | **81.07** | 79.75 | **82.34** | 78.18 | **79.07** |
| WS | 70.31 | **73.02** | 71.52 | **74.65** | 71.19 | **73.06** | 71.5 | **74.78** |
| RW | 45.86 | **48.4** | 44.96 | **49** | 44.44 | **49.22** | 44.5 | **49.03** |
| MEN | 75.84 | **78.21** | 75.84 | **77.96** | 76.16 | **78.35** | 76.72 | **78.1** |
| Mturk | 67.47 | **67.79** | 67.67 | **68** | 67.98 | **68.87** | 68.34 | **69.44** |
| SimLex | 35.3 | **37.59** | 36.54 | **37.85** | 36.62 | **38.44** | 36.67 | **38.58** |
| SimVerb | 22.81 | **25.6** | 23.48 | **25.57** | 23.68 | **25.76** | 23.24 | **26.58** |

Table 15: Before-After results (x100) on word similarity task on seven datasets.

| Dim | 300 | | 400 | | 500 | | 600 | |
|---|---|---|---|---|---|---|---|---|
| | orig. | proc. | orig. | proc. | orig. | proc. | orig. | proc. |
| ap | 46.1 | **48.61** | 42.57 | **45.34** | 46.85 | **50.88** | 40.3 | **45.84** |
| esslli | 68.18 | **72.73** | 64.2 | **82.72** | 64.2 | **65.43** | 65.91 | **72.73** |
| battig | 71.6 | **77.78** | 68.18 | **75** | 68.18 | **70.45** | 46.91 | **66.67** |
| Dim | 700 | | 800 | | 900 | | 1000 | |
| | orig. | proc. | orig. | proc. | orig. | proc. | orig. | proc. |
| ap | 38.04 | **41.31** | 34.76 | **39.8** | **34.76** | 27.46 | 27.96 | **28.21** |
| esslli | 54.55 | 54.55 | **68.18** | 56.82 | 72.73 | 72.73 | 52.27 | 52.27 |
| battig | 62.96 | **66.67** | 67.9 | **69.14** | 49.38 | **59.26** | **51.85** | 46.91 |

Table 16: Before-After results (x100) on the categorization task.

| Dim | 300 | | 400 | | 500 | | 600 | |
|---|---|---|---|---|---|---|---|---|
| | orig. | proc. | orig. | proc. | orig. | proc. | orig. | proc. |
| syn. | 60.48 | **60.52** | **61.61** | 61.45 | **60.93** | 60.84 | **61.66** | 61.57 |
| sem. | 74.51 | **74.54** | 77.11 | **77.36** | 76.39 | **76.89** | 77.28 | **77.61** |
| all. | 66.86 | **66.87** | 68.66 | **68.69** | 67.88 | **68.11** | 68.77 | **68.81** |
| Dim | 700 | | 800 | | 900 | | 1000 | |
| | orig. | proc. | orig. | proc. | orig. | proc. | orig. | proc. |
| syn. | 60.94 | **61.02** | **68.38** | 68.34 | 60.47 | **60.30** | **67.56** | 67.30 |
| sem. | 77.24 | **77.26** | 77.24 | **77.35** | 76.76 | **76.90** | **76.71** | 76.51 |
| all. | 68.36 | **68.41** | 68.38 | **68.50** | **67.91** | 67.67 | **67.56** | 67.30 |

Table 17: Before-After results (x100) on the word analogy task.

## F   STATISTICS OF TEXT CLASSIFICATION DATASETS

We evaluate the word representations (with and without postprocessing) using four different neural network architectures (CNN, vanilla-RNN, GRU-RNN and LSTM-RNN) on five benchmarks:

- the movie review (MR) dataset (Pang & Lee, 2005) where each review is composed by only one sentence;

- the subjectivity (SUBJ) dataset (Pang & Lee, 2004) where the algorithm needs to decide whether a sentence is subjective or objective;

| Dim | 300 | | 400 | | 500 | | 600 | |
|---|---|---|---|---|---|---|---|---|
| | orig. | proc. | orig. | proc. | orig. | proc. | orig. | proc. |
| 2012 | 54.51 | **54.95** | 54.31 | **54.57** | 55.13 | **56.23** | 55.35 | **56.03** |
| 2013 | 56.58 | **57.89** | 56.35 | **57.35** | 57.55 | **59.38** | 57.43 | **59.00** |
| 2014 | 59.6 | **61.92** | 59.57 | **61.62** | 61.19 | **64.38** | 61.10 | **63.86** |
| 2015 | 59.65 | **61.48** | 59.69 | **61.19** | 61.63 | **64.77** | 61.42 | **64.04** |
| SICK | 68.89 | **70.79** | 60.6 | **70.27** | 68.63 | **71.00** | 68.58 | **70.57** |
| all | 58.32 | **59.91** | 58.25 | **59.55** | 59.61 | **62.02** | 59.57 | **61.55** |
| Dim | 700 | | 800 | | 900 | | 1000 | |
| | orig. | proc. | orig. | proc. | orig. | proc. | orig. | proc. |
| 2012 | 55.52 | **56.49** | 54.47 | **54.85** | 54.69 | **55.18** | 54.34 | **54.78** |
| 2013 | 57.61 | **59.31** | 56.75 | **57.62** | 56.98 | **58.26** | 56.78 | **57.73** |
| 2014 | 61.57 | **64.77** | 60.51 | **62.83** | 60.89 | **63.34** | 60.78 | **63.03** |
| 2015 | 62.05 | **65.45** | 60.74 | **62.84** | 61.09 | **63.48** | 60.92 | **63.03** |
| SICK | 68.38 | **70.63** | 67.94 | **69.59** | 67.86 | **69.5** | 67.58 | **69.16** |
| all | 59.96 | **62.34** | 58.87 | **60.39** | 59.16 | **60.88** | 58.94 | **60.48** |

Table 18: Before-After results (x100) on the semantic textual similarity tasks.

| | WORD2VEC | | GLOVE | |
|---|---|---|---|---|
| | orig. | proc. | orig. | proc. |
| capital-common-countries | 82.01 | **83.60** | 95.06 | 95.96 |
| capital-world | 78.38 | **80.08** | 91.89 | **92.31** |
| city-in-state | 69.56 | **69.88** | 69.56 | **70.45** |
| currency | 32.43 | **32.92** | **21.59** | 21.36 |
| family | **84.98** | 84.59 | **95.84** | 95.65 |
| gram1-adjective-to-adverb | **28.02** | 27.72 | **40.42** | 39.21 |
| gram2-opposite | 40.14 | **40.51** | **31.65** | 30.91 |
| gram3-comparative | 89.19 | **89.26** | 86.93 | **87.09** |
| gram4-superlative | 82.71 | **83.33** | 90.46 | **90.59** |
| gram5-present-participle | 79.36 | **79.64** | **82.95** | 82.76 |
| gram6-nationality-adjective | 90.24 | **90.36** | 90.24 | 90.24 |
| gram7-past-tense | 66.03 | **66.53** | 63.91 | **64.87** |
| gram8-plural | 91.07 | 90.61 | 95.27 | **95.36** |
| gram9-plural-verbs | **68.74** | 67.58 | 67.24 | **68.05** |

Table 19: Before-After results (x100) on the word analogy task.

- the TREC question dataset (Li & Roth, 2002) where all the questions in this dataset has to be partitioned into six categories;

- the IMDb dataset (Maas et al., 2011) – each review consists of several sentences;

- the Stanford sentiment treebank (SST) dataset (Socher et al., 2013a), where we only use the full sentences as the training data.

In TREC, SST and IMDb, the datasets have already been split into train/test sets. Otherwise we use 10-fold cross validation in the remaining datasets (i.e., MR and SUBJ). Detailed statistics of various features of each of the datasets are provided in Table 21.

## G  PROOF OF THEOREM A.1

Given the similarity between the setup in Theorem 2.2 in (Arora et al., 2016) and Theorem A.1, many parts of the original proof can be reused except one key aspect – the concentration of $Z(c)$. We summarize this part in the following lemma:

| | WORD2VEC | | GLOVE | |
|---|---|---|---|---|
| | orig. | proc. | orig. | proc. |
| 2012.MSRpar | 42.12 | **43.85** | **44.54** | 44.09 |
| 2012.MSRvid | 72.07 | **72.16** | 64.47 | **68.05** |
| 2012.OnWN | 69.38 | **69.48** | 53.07 | **65.67** |
| 2012.SMTeuroparl | 53.15 | **54.32** | 41.74 | **45.28** |
| 2012.SMTnews | **49.37** | 48.53 | 37.54 | **47.22** |
| 2013.FNWN | 40.70 | **41.96** | 37.54 | **39.34** |
| 2013.OnWN | 67.87 | **68.17** | 47.22 | **58.60** |
| 2013.headlines | 61.88 | **63.81** | 49.73 | **57.20** |
| 2014.OnWN | 74.61 | **74.78** | 57.41 | **67.56** |
| 2014.deft-forum | 32.19 | **33.26** | 21.55 | **29.39** |
| 2014.deft-news | **66.83** | 65.96 | 65.14 | **71.45** |
| 2014.headlines | 58.01 | **59.58** | 47.05 | **52.60** |
| 2014.images | 73.75 | **74.17** | 57.22 | **68.28** |
| 2014.tweet-news | 71.92 | **72.07** | 58.32 | **66.13** |
| 2015.answers-forum | 46.35 | **46.80** | 30.02 | **39.86** |
| 2015.answers-students | **68.07** | 67.99 | 49.20 | **62.38** |
| 2015.belief | 59.72 | **60.42** | 44.05 | **57.68** |
| 2015.headlines | 61.47 | **63.45** | 46.22 | **53.31** |
| 2015.images | 78.09 | **78.08** | 66.63 | **73.20** |
| SICK | 70.10 | **70.20** | 65.14 | **67.85** |

Table 20: Before-After results (x100) on the semantic textual similarity tasks.

| | $c$ | $l$ | Train | Test |
|---|---|---|---|---|
| MR | 2 | 20 | 10,662 | 10-fold cross validation |
| SUBJ | 2 | 23 | 10,000 | 10-fold cross validation |
| TREC | 6 | 10 | 5,952 | 500 |
| SST | 5 | 18 | 11,855 | 2,210 |
| IMDb | 2 | 100 | 25,000 | 25,000 |

Table 21: Statistics for the five datasets after tokenization: $c$ represents the number of classes; $l$ represents the average sentence length; Train represents the size of the training set; and Test represent the size of the test set.

**Lemma G.1** *Let $c$ be a random variable uniformly distributed over the unit sphere, we prove that with high probability, $Z(c)/|\mathcal{V}|$ converges to a constant $Z$:*

$$p((1 - \epsilon_z)Z \leq Z(c) \leq (1 + \epsilon_z)Z) \geq 1 - \delta,$$

*where $\epsilon_z = \Omega((D + 1)/|\mathcal{V}|)$ and $\delta = \Omega((DA^2 + \|\mu\|^2)/d)$.*

Our proof differs from the one in (Arora et al., 2016) in two ways: (a) we treat $v(w)$ as deterministic parameters instead of random variables and prove the Lemma by showing a certain concentration of measure; (b) the asymmetric parts $\mu$ and $u_1, ..., u_D$, (which did not exist in the original proof), need to be carefully addressed to complete the proof.

### G.1 PROOF OF LEMMA G.1

Given the constraints on the word vectors (3), the partition function $Z(c)$ can be rewritten as,

$$Z(c) = \sum_{v \in \mathcal{V}} \exp(c^\top v(w))$$

$$= \sum_{v \in \mathcal{V}} \exp\left( c^\top \left( \mu + \sum_{i=1}^{D} \alpha_i(w)u_i + \tilde{v}(w) \right) \right)$$

$$= \sum_{v \in \mathcal{V}} \exp(c^\top \mu) \left[ \prod_{i=1}^{D} \exp(\alpha_i(w)c^\top u_i) \right] \exp\left( c^\top \tilde{v}(w) \right).$$

The equation above suggests that we can divide the proof into five parts.

**Step 1:** for every unit vector $c$, one has,

$$\frac{1}{|\mathcal{V}|} \sum_{w \in \mathcal{V}} \exp\left( c^\top \tilde{v}(w) \right) \to \mathbb{E}_f \left( \exp\left( c^\top \tilde{v} \right) \right), \text{ as } |\mathcal{V}| \to \infty. \tag{5}$$

**Proof** Let $M, N$ be a positive integer, and let $A_M \subset \mathbb{R}^d$ such that,

$$A_{M,N} = \left\{ \tilde{v} \in \mathbb{R}^d : \frac{M-1}{N} < \exp(c^\top \tilde{v}) \leq \frac{M}{N} \right\}.$$

Since $A_{M,N}$ can be represented by a union of countable disjoint rectangles, we know that for every $M, N \in \mathbb{N}_+$,

$$\frac{1}{|\mathcal{V}|} \sum_{w \in \mathcal{V}} \mathbf{1}(\tilde{v}(w) \in A_{M,N}) = \int_{A_{M,N}} f(\tilde{v})d\tilde{v}.$$

Further, since $A_{M,N}$ are disjoint for different $M$'s and $\mathbb{R}^d = \cup_{M=1}^{\infty} A_{M,N}$, one has,

$$\frac{1}{|\mathcal{V}|} \sum_{w \in \mathcal{V}} \exp\left( c^\top \tilde{v}(w) \right) = \sum_{M=1}^{\infty} \frac{1}{|\mathcal{V}|} \sum_{w \in \mathcal{V}} \mathbf{1}(\tilde{v}(w) \in A_{M,N}) \exp(c^\top \tilde{v}(w))$$

$$\leq \sum_{M=1}^{\infty} \frac{1}{|\mathcal{V}|} \sum_{w \in \mathcal{V}} \mathbf{1}(\tilde{v}(w) \in A_{M,N}) \frac{M}{N}$$

$$\to \sum_{M=1}^{\infty} \frac{M}{N} \int_{A_{M,N}} f(\tilde{v})d\tilde{v}.$$

The above statement holds for every $N$. Let $N \to \infty$, by definition of integration, one has,

$$\lim_{N \to \infty} \sum_{M=1}^{\infty} \frac{M}{N} \int_{A_{M,N}} f(\tilde{v})d\tilde{v} = \mathbb{E}_f \left( \exp\left( c^\top \tilde{v} \right) \right),$$

which yields,

$$\frac{1}{|\mathcal{V}|} \sum_{w \in \mathcal{V}} \exp\left( c^\top \tilde{v}(w) \right) \leq \mathbb{E}_f \left( \exp\left( c^\top \tilde{v} \right) \right), \text{ as } |\mathcal{V}| \to \infty. \tag{6}$$

Similarly, one has,

$$\frac{1}{|\mathcal{V}|} \sum_{w \in \mathcal{V}} \exp\left( c^\top \tilde{v}(w) \right) \geq \lim_{N \to \infty} \sum_{M=1}^{\infty} \frac{M-1}{N} \int_{A_{M,N}} f(\tilde{v})d\tilde{v}$$

$$= \mathbb{E}_f \left( \exp\left( c^\top \tilde{v} \right) \right), \text{ as } |\mathcal{V}| \to \infty. \tag{7}$$

Putting (6) and (7) proves (5).

**Step 2:** the expected value, $\mathbb{E}_f\left(\exp\left(c^\top \tilde{v}\right)\right)$ is a constant independent of $c$:

$$\mathbb{E}_f\left(\exp\left(c^\top \tilde{v}\right)\right) = Z_0. \tag{8}$$

**Proof** Let $Q \in \mathbb{R}^{d\times d}$ be a orthonormal matrix such that $Q^\top c_0 = c$ where $c_0 = (1,0,...,0)^\top$ and $\det(Q) = 1$, then we have $f(\tilde{v}) = f(Q\tilde{v})$, and,

$$\begin{aligned}
\mathbb{E}_f\left(\exp\left(c_0^\top \tilde{v}\right)\right) &= \int_{\tilde{v}} f(\tilde{v}) \exp\left(c_0^\top \tilde{v}\right) d\tilde{v} \\
&= \int_{\tilde{v}} f(Q\tilde{v}) \exp\left(c^\top Q\tilde{v}\right) \det(Q) d\tilde{v} \\
&= \int_{\tilde{v}'} f(\tilde{v}') \exp\left(c^\top \tilde{v}'\right) d\tilde{v}' = \mathbb{E}_f\left(\exp\left(c^\top \tilde{v}\right)\right),
\end{aligned}$$

which proves (8).

**Step 3:** for any vector $\mu$, one has the following concentration property,

$$p\left(|\exp\left(c^\top \mu\right) - 1| > k\right) \le 2\left[\exp\left(-\frac{1}{4}\right) + \frac{\|\mu\|^2}{d-1}\frac{1}{\log^2(1-k)}\right] \tag{9}$$

**Proof** Let $c_1,...,c_d$ be i.i.d. $\mathcal{N}(0,1)$, and let $C = \sum_{i=1}^d c_i^2$, then $c = (c_1,...,c_d)/\sqrt{C}$ is uniform over unit sphere. Since $c$ is uniform, then without loss of generality we can consider $\mu = (\|\mu\|, 0, ..., 0)$. Thus it suffices to bound $\exp\left(\|\mu\|c_1/\sqrt{C}\right)$. We divide the proof into the following steps:

- $C$ follows chi-square distribution with the degree of freedom of $d$, thus $C$ can be bounded by (Laurent & Massart, 2000),

$$p(C \ge d + 2\sqrt{dx} + 2x) \le \exp(-x), \forall x > 0. \tag{10}$$
$$p(C \le d - 2\sqrt{dx}) \le \exp(-x), \forall x > 0. \tag{11}$$

- Therefore for any $x > 0$, one has,

$$p\left(|C - d| \ge 2\sqrt{dx}\right) \le \exp(-x)$$

Let $x = 1/4d$, one has,

$$p(C > d + 1) \le \exp\left(-\frac{1}{4d}\right),$$
$$p(C < d - 1) \le \exp\left(-\frac{1}{4d}\right).$$

- Since $c_1$ is a Gaussian random variable with variance 1, by Chebyshev's inequality, one has,

$$p\left(yc_i \ge k\right) \le \frac{y^2}{k^2}, \qquad p\left(yc_i \le -k\right) \le \frac{y^2}{k^2}, \forall k > 0$$

and therefore thus,

$$p(\exp(yc_i) - 1 > k) \le \frac{y^2}{\log^2(1+k)},$$
$$p(\exp(yc_i) - 1 < -k) \le \frac{y^2}{\log(1-k)^2}, \forall k > 0.$$

- Therefore we can bound $\exp\left(\|\mu\|c_1/\sqrt{C}\right)$ by,

$$p\left(\exp\left(\frac{\|\mu\|c_1}{\sqrt{C}}\right) - 1 > k\right) \leq p\left(C > d+1\right)$$

$$+ p\left(\exp\left(\frac{\|\mu\|c_1}{\sqrt{C}}\right) - 1 > k \middle| C < d+1\right) p\left(C < d+1\right)$$

$$\leq \exp\left(-\frac{1}{4d}\right) + p\left(\exp\left(\frac{\|\mu\|c_1}{\sqrt{d+1}}\right) - 1 > k\right)$$

$$= \exp\left(-\frac{1}{4d}\right) + \frac{\|\mu\|^2}{d+1} \frac{1}{\log(1-k)^2}.$$

$$p\left(\exp\left(\frac{\|\mu\|c_1}{\sqrt{C}}\right) - 1 < -k\right) \leq \exp\left(-\frac{1}{4d}\right) + \frac{\|\mu\|^2}{d-1} \frac{1}{\log^2(1+k)}.$$

Combining the two inequalities above, one has (9) proved.

**Step 4:** We are now ready to prove convergence of $Z(c)$. With (9), let $\mathcal{C} \subset \mathbb{R}^d$ such that,

$$\mathcal{C} = \left\{c : \left|\exp(c^\top \mu) - 1\right| < k, \left|\exp(Ac^\top u_i) - 1\right| < k, \left|\exp(-Ac^\top u_i) - 1\right| < k \ \forall i = 1, ..., D\right\}$$

Then we can bound the probability on $\mathcal{C}$ by,

$$p(\mathcal{C}) \geq p\left(\left|\exp(c^\top \mu) - 1\right| < k\right) + \sum_{i=1}^{D} p\left(\left|\exp(Ac^\top u_i) - 1\right| < k\right) - 2D$$

$$\geq 1 - (2D+1)\exp\left(-\frac{1}{4d}\right) - \frac{2DA^2}{d-1}\frac{1}{\log^2(1-k)} - \frac{\|\mu\|^2}{d-1}\frac{1}{\log^2(1-k)}.$$

Next, we need to show that for every $w$, the corresponding $\mathcal{C}(w)$, i.e.,

$$\mathcal{C}(w) = \left\{c : \left|\exp(c^\top \mu) - 1\right| < k, \left|\exp(\alpha_i(w)c^\top u_i) - 1\right| < k, \ \forall i = 1, ..., D\right\}$$

We observe that $\alpha_i(w)$ is bounded by $A$, therefore for any $c$ that,

$$\min(\exp(-Ac^\top u_i), \exp(Ac^\top u_i)) \leq \exp(\alpha_i c^\top u_i) \leq \max(\exp(-Ac^\top u_i), \exp(Ac^\top u_i)),$$

and thus,

$$\min(\exp(-Ac^\top u_i), \exp(Ac^\top u_i)) - 1 \leq \exp(\alpha_i c^\top u_i) - 1 \leq \max(\exp(-Ac^\top u_i), \exp(Ac^\top u_i)) - 1,$$

which yields,

$$|\exp(\alpha_i c^\top u_i) - 1| \leq \max(|\exp(-Ac^\top u_i) - 1|, |\exp(Ac^\top u_i) - 1|) < k.$$

Therefore we prove $\mathcal{C}(w) \supset \mathcal{C}$. Assembling everything together, one has,

$$p\left(\left|\exp(c^\top \mu)\prod_{i=1}^{D}\exp(\alpha_i(w)c^\top u_i) - 1\right| > (D+1)k, \ \forall i = 1, ..., D, \forall w \in \mathcal{V}\right)$$

$$\leq p(\bar{\mathcal{C}})$$

$$\leq (2D+1)\exp(-\frac{1}{4d}) + \frac{2DA^2}{d-1}\frac{1}{\log^2(1-k)} + \frac{\|\mu\|^2}{d-1}\frac{1}{\log^2(1-k)}$$

For every $c \in \mathcal{C}$, one has,

$$\frac{1}{|\mathcal{V}|}|Z(c) - Z_0| \leq \frac{(D+1)k}{|\mathcal{V}|}Z_0.$$

Let $Z = |\mathcal{V}|Z_0$, one can conclude that,

$$p((1-\epsilon_z)Z \leq Z(c) \leq (1+\epsilon_z)Z) \geq 1 - \delta,$$

where $\epsilon_z = \Omega((D+1)/|\mathcal{V}|)$ and $\delta = \Omega(DA^2/d)$.

## G.2   PROOF OF THEOREM A.1

Having Lemma G.1 ready, we can follow the same proof as in (Arora et al., 2016) that both $p(w)$ and $p(w, w')$ are correlated with $\|v(w)\|$, formally

$$\log p(w) \to \frac{\|v(w)\|^2}{2d} - \log Z, \text{ as } |\mathcal{V}| \to \infty, \tag{12}$$

$$\log p(w, w') \to \frac{\|v(w) + v(w')\|^2}{2d} - \log Z, \text{ as } |\mathcal{V}| \to \infty. \tag{13}$$

Therefore, the inference presented in (Arora et al., 2016) (i.e., (4)) is obvious by assembling (12) and (13) together:

$$\mathrm{PMI}(w, w') \to \frac{v(w)^\top v(w')}{d}, \text{ as } |\mathcal{V}| \to \infty.$$

