# OpenReview forum: "All-but-the-Top: Simple and Effective Postprocessing for Word Representations"
_ICLR.cc/2018/Conference — Accept (Poster)_

### Official Review · AnonReviewer3 · 2017-11-27
**In-depth explanation of known phenomenon. Experiments need some strengthening.**

**Rating:** 6
**Confidence:** 5

**Review:**

This paper provides theoretical and empirical motivations for removing the top few principle components of commonly-used word embeddings.

The paper is well-written and I enjoyed reading it. However, it does not explain how significant this result is beyond that of (Bullinaria and Levy, 2012), who also removed the top N dimensions when benchmarking SVD-factorized word embeddings. From what I can see, this paper provides a more detailed explanation of the phenomenon ("why" it works), which is supported with both theoretical results and a series of empirical analyses, as well as "updating" the benchmarks and methods from the pre-neural era. Although this contribution is relatively incremental, I find the depth of this work very interesting, and I think future work could perhaps rely on these insights to create better embedding algorithms that directly enforce isotropy.

I have two concerns regarding the empirical section, which may be resolvable fairly quickly:
1) Are the embedding vectors L2 normalized before using them in each task? This is known to significantly affect performance. I am curious whether removing the top PCs is redundant or not given L2 normalization.
2) Most of the benchmarks used in this paper are "toy" tasks. As Schnabel et al (2015) and Tsvetkov et al (2015) showed, there is often little correlation between success on these benchmarks and improvement of downstream NLP tasks. I would like to measure the change in performance on a major NLP task that heavily relies on pre-trained word embeddings such as SQuAD.

Minor Comments:
* The last sentence in the first paragraph ("The success comes from the geometry of the representations...") is not true; the success stems from the ability to capture lexical similarity. Levy and Goldberg (2014) showed that searching for the closest word vector to (king - man + woman) is equivalent to optimizing a linear combination of 3 similarity terms [+(x,king), -(x,man), +(x, woman)]. This explanation was further demonstrated by Linzen (2016) who showed that even when removing the negative term (x, man), many analogies can still be solved, i.e. by looking for a word that is similar both to "king" and to "woman". Add to that the fact that the analogy trick works best when the vectors are L2 normalized; if they are all on the unit sphere, what is the geometric interpretation of (king - man + woman), which is not on the unit sphere? I suggest removing this sentence and other references to linguistic regularities from this paper, since they are controversial at best, and distract from the main findings.
* This is also related to Bullinaria and Levy's (2012) finding that downweighting the eigenvalue matrix in SVD-based methods improves their performance. Levy et al (2015) showed that keeping the original eigenvalues can actually degenerate SVD-based embeddings. Perhaps there is a connection to the findings in this paper?

---

> ### Author Response · Authors · 2017-12-16
> **Response to AnonReviewer3**
>
> Dear AnonReviewer3,
>
> Thanks for your comments!
>
> Q: Connecting to Bullinaria and Levy's (2012) and Levy et al (2015):
> A: Thanks for pointing out these references. The key difference between our approach and the previous work is that we null out the top PCs of the *low-dimensional* word vectors (as the factorization of the cooccurrence matrix) while the previous works downgrade or remove the top PCs of the cooccurrence matrix itself. Specifically in the example of equation (5) and (6) in Levy et al (2015), downgrading the top PCs means downgrading the first few *dimensions* of the word vectors, not downgrading the top *PCs*.
>
> Q: Are the embedding vectors L2 normalized before using them in each task? This is known to significantly affect performance. I am curious whether removing the top PCs is redundant or not given L2 normalization:
> A: We also experimented with L2 normalized vectors — the results are similar, i.e., there is also a consistent improvement on all benchmarks in this paper. We are able to explain this phenomenon as follows: the word vectors almost all have the same norm; example: for  GLOVE (average norm: 8.30, std: 1.56).  Therefore normalizing it or not barely make any impact on the directions of top PCs and therefore does not affect the performances of the post processing. However, L2 normalization might affect the SVD-based vectors (for example Levy et al (2015)).
>
> Q: Impact on major NLP tasks:
> A: Showing that the post processing does not only work for the ``toy’’ examples but also in the real applications is exactly what we tried to do with the semantic textual similarity in Section 3 and the text classification task in Section 4. We are aware that there are other sophisticated downstream applications involving complicated neural networks — Q&A in SQuAD and even machine translation in WMT. Apart from the fact that we don’t have access to the source code to the top systems presented in the leaderboard, doing these experiments would take huge resources in terms of both timing and computational resources. We submit that it’s hard to exhaust all downstream applications in this one submission with our individual effort. Having said this, if the reviewer has a specific downstream application in mind and a specific code to suggest we are happy to experiment with it.
>
> Q: Linguistic regularities.
> A: Thanks for your advice. We have removed this part from our paper.

---

### Official Review · AnonReviewer1 · 2017-11-28
**Simple method to tidy up word embedding spaces; novelty unclear, conclusions opaque**

**Rating:** 7
**Confidence:** 4

**Review:**

This paper proposes that sets of word embeddings can be improved by subtracting the common mean vector and reducing the effect of dominant components of variation.

Comments:

Reference to 'energy' and 'isotropic' in the first paragraph come without any explanation. Can plain terms be used instead to express the same ideas? This would make understanding easier (I have a degree in maths, but never studied physics, and I had to look them up). Otherwise, I like the simple explanation of the method given in the intro.

The experiments conducted in the paper are comprehensive. It is very positive that the improvements appear to be quite consistent across well-known tasks. As well as proposing a simple trick to produce these improvements, the authors aim to provide theoretical insight and (implicitly, at least) pursue a better understanding of semantic word spaces. This has the potential to be a major contribution, as such spaces, and semantic representation in neural nets in general, is poorly understood. However (perhaps because I'm not familiar with Arora et al.) I found the mathematical analysis e.g. S2.2 dense, without any clearly-stated intuitive motivation or conclusions (as per the introduction section) about what is going on semantically. E.g. it is not clear to me why isotropy is something desirable in a word embedding space. I understand that the discarded components tend to encode frequency, and this is very interesting and somewhat indicative f why the method might work. However, Figure 2 is particularly hard to interpret? The correlations, and the distribution of high-frequency words) seems to be quite different for each of the three models?!

In general, I don't think the authors should rely on readers having read Arora et al. - anything that builds on that work needs to reintroduce their findings in pain terms in the current paper.

Another concern is the novelty in relation to related work. I have not read Arora et al. but the authors say that they 'null away the first principal component', and Sahlgren et al centre the mean. Taken together, this seems very similar to what the authors propose here (please clarify). More generally, these sorts of tricks have often been applied by deep learning researchers and passed around anecdotally (e.g. initialise transition matrix in RNNs with orthonormal noise) as ways to improve training. It is important to share and verify these things, but such a contribution feels more appropriate for a workshop than the main conference. This makes the work that the authors do in interpreting and understanding why these tricks work particularly important. As is, however, I thing that the conclusions from this analysis are unclear and opaque. Can they be better communicated, or is it the case that the results of the analysis are in fact inconclusive?

The vast amount of work included in the appendix is impressive. What particularly caught my eye was appendix B, where the authors try to understand if their method can simply be 'learned' by any network that uses pre-trained word embeddings. This is a really nice experiment, and I think it could easily be part of the main paper (perhaps swapping with the stuff in section 2.2).

The conclusion would be a good place for summarising the main findings in plain terms, but that doesn't really happen (unless the finding about frequency is the only finding). Instead, there is a vague connection to population genetics and language evolution. This may be an interesting future direction, but the connection is tenuous, so that this reader, at least, was left a little baffled.

[REVISED following response]

Thanks for your thorough response which did a good job of addressing most of my concerns. I have changed the score accordingly.

---

> ### Author Response · Authors · 2017-12-16
> **Response to AnonReviewer1**
>
> Dear AnonReviewer1,
>
> Thanks for your comments.
>
> Q: Why isotropy is desirable in a word embedding space.
> A: A priori, there is no scientific reason for isotropy (other than the heuristic idea that isotropy spreads out the word vectors and the larger separation potentially leads to better representations). The isotropic property was introduced in (Arora et al. 2015) as an axiom, but primarily their theorem on self-normalization to go through. Motivated by this work, we ask if explicitly imposing the isotropy constraint we can get better empirical performance -- the results in this paper answer this affirmatively.
>
> Q: Interpretation of Figure 2.
> A: Our main point behind Figure 2 is the finding that the top coefficients capture the frequency to some extent -- we couldnt quantify the extent in a precise manner; any suggestions are welcome. Also note that frequency may not be the only aspect capture in the top coefficients.
> Re correlations being different in the three models:  this statement is not exactly true, for two reasons. First, the coefficients are equivalent up to a +/- sign, this is because singular vectors by themselves do not capture the directions, and therefore the correlations in CBOW and SKIP-GRAM are quite similar (they are reflectional symmetric w.r.t. the y-axis). Second, the training algorithms for GloVe (as a factorization of the co-occurrence matrix) and word2vec (directly optimizing parameters in a probabilistic model) are rather different. Even though a lot of works try to tighten them together, there is no conclusion that these two methods produce the same output.
>
> Q: Prior work on (Arora et al, 2015):
> A: We have made our manuscript as much self contained as possible (with small additions to enable this in the newly revised version, cf. top of page 4). Indeed, the main part o the paper barely requires the reader to know the results of (Arora et al, 2015).  If the reviewer can suggest specific places where we could improve our exposition, that would be very much appreciated.  The mathematical connections between our work and RAND-WALK are explored in detail in Appendix A and have made this material self contained . Given size limitations of the conference paper, we chose to emphasize our post-processing algorithm and its empirical performances in the main text and move these mathematical connections to the appendix.
>
> Q: Novelty in relation to the related work.
> A: We first note that our post-processing algorithm was derived independently of the works of (Arora et al. (2017)) and (Sahlgren et al. (2016)) --  time-stamps document this fact but we cannot divulge explicitly due to author anonymity. Second, although there is a superficial similarity between our work and (Arora et al. 2017), the nulling directions we take and the one they take are fundamentally different. Specifically, in Arora et al. (2017), the first dominating vector is *dataset-specific*, i.e., the first compute the sentence representation for the entire STS dataset, then extract the top direction from those sentence representations and finally project the sentence representation away from it. By doing so, the top direction will inherently encode the common information across the entire dataset, the top direction for the "headlines" dataset may encode common information about news articles while the top direction for "Twitter'15" may encode the common information about tweets. In contrast, our dominating vectors are over the entire vocabulary of the language.
>
> We also updated the related work section in the newly revised version to highlight these  differences.
>
> Q: Explanation and interpretation of these tricks.
> A: The "simple tricks" that are originally found to work empirically very well, have  a mathematical basis underlying them on several occasions. Indeed, the "trick" mentioned by the reviewer -- initializing transition matrix in RNNs with orthonormal matrix -- is a paper at ICML 2017 (Vorontsov et al. 2017).  We have proposed post-processing as a good initialization on word representations in downstream tasks. We submit that our paper makes multi-faceted attempts to explain our initialization procedure: (a) by theoretically analyzing the isotropy property of word representations; (b)  empirically demonstrating the  boosted performances of word representations consistently in various tasks; (c) a  study on the connection between the postprocessing algorithms and the arithmetic operations learnt end-to-end by modern day deep learning NLP pipelines.
>
> Q: Conclusion.
> Thanks for the advice; we appreciate it.  We have updated the conclusion and summarize the main point of our paper: the simple post-processing operation should be used for word embeddings in downstream tasks or as intializations for training task-specific embeddings.
>
> Thanks!

---

### Official Review · AnonReviewer2 · 2017-11-29
**Simple post-processing technique with theoretical motivations**

**Rating:** 7
**Confidence:** 4

**Review:**

This paper proposes a simple post-processing technique for word representations designed to improve representational quality and performance on downstream tasks. The procedure involves mean subtraction followed by projecting out the first D principle directions and is motivated by improving isotropy of the partition function. Extensive empirical analysis supports the efficacy of the approach.

The idea of post-processing word embeddings to improve their performance is not new, but I believe the specific procedure and its connection to the concept of isotropy has not been investigated previously. Relative to other post-processing techniques, this method has a fair amount of theoretical justification, particularly as described in Appendix A. I think the experiments are reasonably comprehensive. All told, I think this is a good paper, but I do have some comments and questions that I think should be addressed before publication.

1) I think it is useful to analyze the distribution of singular values of the matrix of word vectors. However, I did not find the heuristic analysis based on the visual appearance of these distributions to be convincing. For example, in Fig. 1, it is not clear to me that there exists a separation between regimes of exponential decay and rough constancy. It would be ideal if a more quantitative metric is established that captures the main qualitative behavior alluded to here.

Furthermore, the vocabulary size is likely to have a strong effect on the shape of the distributions. Are the plots in Fig. 4 for the same vocabulary size? Related to this, the dimensionality of the representation will have a strong effect on the shape, and this should be controlled for in Fig. 8. One way to do this would be to instead plot the density of singular values. Finally, for the Gaussian matrix simulations, in the asymptotic limit, the density of singular values depends only on the ratio of dimensions, i.e. the vector dimension to the vocabulary size. Fig. 4/8 might be more revealing if this ratio were controlled for.

2) It would be useful to describe why isotropy of the partition function is the goal, as opposed to isotropy of the vectors themselves. This may be argued in Arora et al. (2016), but summarizing that argument in this paper would be helpful. In fact, an additional experiment that would be very valuable would be to investigate empirically which form of isotropy is more effective in governing performance. One way to do this would be to enforce approximate isotropy of the partition function without also enforcing isotropy of the vectors themselves. Practically speaking, one might imagine doing this by requiring I = 1 to second order without also requiring that the mean vanish. I think this would allow for \sigma_max > \sigma_min while still satisfying I = 1 to second order. (But this is just off the top of my head -- there may be better ways to conduct this experiment).

It is not clear to me why the experiment leading to Table 2 is a good proxy for the exact computation of I. It would be great if there were some mathematical justification for this approximation.

Why does Fig. 3 use D=10, 20 when much smaller D are considered elsewhere? Also I think a log scale on the x-axis might be more informative.

3) It would be good to mention other forms of post-processing, especially in the context of word similarity. For example, in the original paper, GloVe advocates averaging the target and context vector representations, and normalizing across the feature dimension before computing cosine similarity.

4) I think it's likely that there is a strong connection between the optimal value of D and the frequency distribution of words in the evaluation dataset. While the paper does mention that D may depend on specifics of the dataset, etc., I would expect frequency-dependence to be the main factor, and it might be worth exploring this effect explicitly.

---

> ### Author Response · Authors · 2017-12-16
> **Response to AnonReviewer2**
>
> Dear AnonReviewer2,
>
> Thanks for your comments!
>
> Q: Quantitative metric in Figure 1.
> A: We admit that it's hard to see the separation between two regimes in a log-scale of the x-axis, but we prefer to plot it  this way. This is because other than the top 10 components, the rest of the singular values are very small and almost the same as each other  (as shown in Figure 4) and these constitute a large portion (~95%) of all singular values. We don't want to spend a large portion of the graph capturing this fact. Additionally, in this scale, it's easy to see the decay of the top singular values (approximately exponentially w.r.t. the index).
>
> For completeness, p we plot the singular values in a linear scale of its index in this anonymized link: https://www.dropbox.com/s/marzc41z2oy6qau/decay.pdf?dl=0 . The decay of the eigenvalues is much more obvious in this plot and there is a very clear separation between the two regimes. To complete this discussion, we provide the following table of the normalized singular values (x100) of the first 10 out of 300 components.
>
> GLOVE     | 2.12    1.27    1.11    0.95    0.69    0.69    0.64    0.62    0.60    0.56    0.53    0.52    0.51    0.50    0.49    0.49    0.46    0.45    0.44    0.44
> RAND-WALK | 3.29    2.45    1.89    1.50    1.38    1.21    1.12    1.03    0.96    0.89    0.84    0.79    0.77    0.69    0.67    0.64    0.63    0.62    0.60    0.57
> WORD2VEC  | 4.17    2.40    2.23    2.15    1.90    1.61    1.51    1.39    1.30    1.23    1.16    1.06    1.00    0.97    0.95    0.91    0.86    0.85    0.78    0.77
>
> Q: The density of singular values depends on the shape of the distribution.
> A: This is a very subtle point and we really appreciate it being brought out. However, this dependence mentioned by the reviewer is moot in this case for the following reason:  in random matrix theory, the asymptotic density of singular values  depends heavily on the ratio of the size of the matrix, where the asymptotic limit is studied when both dimensions go to infinity (while keeping the ratio unchanged). Such a case does not fit the scenario here: the dimension of word vectors is only 300 which is extremely small with respect to the vocabulary size (~1,000,000).
> In addition, when we plot the figures, we actually take all words in the vocabulary (where the vocabulary size is presented in Table 1). Although different publicly available embeddings are trained on different corpora, the vocabularies are not very different especialy when compared to the size of the word vector dimensions. For instance, Glove and word2vec have essentially the same vocabulary size (~1,000,000).   We dont expect these small differences to  signficantly affect the density of singular vectors, especially when the vocabular sizes are so large compared to the dimension of the word vectors.
>
> Q: Why isotropy of the partition function is the goal, as opposed to the isotropy of the vectors themselves.
> A: Mathematically, istotropy is an asymptotic property of a sequence of vectors. For random vectors, the joint density of an isotropic random vector only depends on the norm of its argument (regardless of the angle). For an infinite sequnence of deterministic vectors, the empirical density of the vectors again is angularly invariant.
>
> For deterministic and finite set of word vectors, there is no one single definition of isotropy. Indeed, one of our contributions is to postulate a ``correct" definition of isotropy (via the partition function) -- this choice allows us to derive practical word embedding algorithms and we empirically demonstrate the improved quality of the resulting word vectors in our paper.
>
> Q: Why D=10,20 is much larger than the D applied elsewhere
> A: Thanks for pointing this out. This is actually a typo; we had chosen D consistently as before (i.e., D = 2 or GloVe and 3 for word2vec). We have fixed this in the revised version.
>
> Q: Other forms of postprocessing
> A: Thank you for this very fair advice. There are various ways to merge two sets of representations (word vectors and the context vectors), taking the average or concatenating them together to form longer vectors are two of the approaches. Several methods we attempted did not yield any noticeable and consistent improvements; example: one way to achieve isotropy is by "whitening the spectrum" which didnt work.  The same holds for context vectors;  if the reviewer has  any specific postprocessing baselines that operate solely on word vectors to suggest, we  are happy to conduct the corresponding experiments.
>
> Thank you.

---

### Decision · Program_Chairs · 2018-01-29
**ICLR 2018 Conference Acceptance Decision**

**Decision:**

Accept (Poster)

**Comment:**

This is a good paper with strong results via a set of simple steps for post processing off the shelf words  embeddings.  Reviewers are enthusiastic about it and the author responses are satisfactory.